

# Inhibitory effect and mechanism of action (MOA) of hirsutine on the proliferation of T-cell leukemia Jurkat clone E6-1 cells

Jie Meng[1,2], Rui Su[1], Luping Wang[1], Bo Yuan[1] and Ling Li[1]

[1] Department of Pharmacy, Tongren Hospital, Shanghai Jiaotong University School of Medicine, Shanghai, China
[2] Hongqiao International Institute of Medicine, Shanghai Jiaotong University School of Medicine, Shanghai, China

Corresponding authors
Bo Yuan, YB2718@shtrhospital.com
Ling Li, LL2699@shtrhospital.com

## ABSTRACT

**Background:** The bark of *Uncaria rhynchophylla* has been traditionally used to treat convulsion, bleeding, hypertension, auto-immune conditions, cancer, and other diseases. The main focus of this research is done for the purpose of exploring the antitumor activity and mechanism of action (MOA) for hirsutine isolated from *U. rhynchophylla*.

**Methods:** Jurkat clone E6-1 cells were treated using 10, 25 and 50 μM for 48 h. Inhibition of cell proliferation due to hirsutine treatment was evaluated by CCK8 assay. Flow cytometry was applied to ascertain Jurkat cell cycle progression and apoptosis after treatment with 10, 25 and 50 μM hirsutine for 48 h. The expression and level of the apoptosis-related genes and proteins was analyzed by Real-time Quantitative polymerase chain reaction (qPCR) and Western blotting method, respectively.

**Results:** CCK8 analyses revealed that hirsutine could significantly inhibit the proliferation of Jurkat clone E6-1 cells, in a concentration and time-dependent fashion. Flow cytometry assays revealed that hirsutine could drive apoptotic death and G0/G1 phase arrest in Jurkat cells. Apoptotic cells frequencies were $4.99 \pm 0.51\%$, $13.69 \pm 2.00\%$ and $40.21 \pm 15.19\%$, and respective cell cycle arrest in G0/G1 accounted for $34.85 \pm 1.81\%$, $42.83 \pm 0.70\%$ and $49.12 \pm 4.07\%$. Simultaneously, compared with the control group, Western blot assays indicated that the up-regulation of pro-apoptotic Bax, cleaved-caspase3, cleaved-caspase9 and Cyto c proteins, as well as the down-regulation of Bcl-2 protein which guards against cell death, might be correlated with cell death induction and inhibition of cell proliferation. QPCR analyses indicated that hirsutine could diminish *BCL2* expression and, at the same time, improve Bax, caspase-3 and caspase-9 mRNA levels, thus reiterating a putative correlation of hirsutine treatment in vitro with apoptosis induction and inhibition of cell proliferation ($p$-value < 0.05). Excessive hirsutine damages the ultrastructure in mitochondria, leading to the release of Cyt c from the mitochondria to cytoplasm in Jurkat clone E6-1 cells, thereby inducing the activated caspase cascade apoptosis process through a mitochondria-mediated pathway.

**Conclusion:** An important bioactive constituent—hirsutine—appears to have antitumor effects in human T-cell leukemia, thus enlightening the use of

phytomedicines as a novel source for tumor therapy. It is speculated that hirsutine may induce apoptosis of Jurkat Clone E6-1 cells through the mitochondrial apoptotic pathway.

## INTRODUCTION

In the past few decades, human T-cell leukemia has become a commonly found malignancy in humans (*Bray et al., 2018*; *Miller et al., 2019*). Currently, human T-cell leukemia is not only an important cause of cancer-related mortality in third world countries, but among the major factors of death in developed countries. The burden of human T-cell leukemia in developing countries have been increasing not only due to population growth and aging, but also due to the lack of exercise, smoking and unhealthy lifestyle (*Reiner et al., 2019*; *Foerster et al., 2018*; *Nunez et al., 2018*; *Yancik & Ries, 2004*). Fortunately, a number of treatments, including surgery (*Li et al., 2019*), radiotherapy (*Mondini et al., 2020*), chemotherapy (*Almodovar et al., 2019*), and traditional Chinese medicine therapy (*Hung et al., 2017*), have been successful to relieve the pain and prolong the life expectancy of patients affected by T-cell leukemia.

Chemicals extracted from some plants are an important source of research into innovative cancer treatments. They have the potential to be highly effective and not overly toxic like other chemicals (*Crowell, 2005*), and many of the chemicals in plants are already being used in health care to help develop new drugs (*Chirumbolo, 2012*; *Uramova et al., 2018*). Hirsutine (Fig. 1) is an indole alkaloid, originally isolated from *Uncaria rhynchophylla*, that has attracted attention on accounts of its biological characteristics in many aspects, like cardioprotective, antihypertensive and antiarrhythmic activities (*Wu et al., 2011*; *Zhu et al., 2015*).

Hirsutine, as a natural drug with less side effects and low toxicity, can be used in many diseases (*Horie et al., 1992*; *Jung et al., 2013*; *Lou et al., 2014*; *Zhang et al., 2018*; *Zhu et al., 2015*). There are now a number of trials showing that hirsutine could be applied as a common drug, besides, it also showed excellent anti-cancer effects in many cell models. Hirsutine is capable of inducing apoptosis inhibition of the HER2, NF-κB and Akt pathways and p38 MAPK cascade activation in several human breast cancer cell lines and appears to be linked with hirsutine-induced DNA damage and apoptosis (*Lou et al., 2015*). It is known that human breast cancer cells treated with hirsutine (i.e., MDA/MB-231) are prone to release mitochondrial cytochrome c and reduce mitochondrial membrane potential (MMP), thus leading to cell apoptosis (*Huang et al., 2018*). Furthermore, hirsutine has been said to potentially inhibits the metastatic features of 4T1 breast cancer cells both in vitro and in vivo by abrogating NF-κB signaling (*Lou et al., 2014*). However, the effects of hirsutine towards apoptosis induction of T-cell leukemia Jurkat cells, and the underlying mechanisms, remain to be elucidated.

**Figure 1 Chemical structure of hirsutine (MW = 368.47).**

Our present work aims to clarify the antiproliferative effects of hirsutine in different human cancer cell lines. Its mechanism of action was particularly evaluated in leukemia cancer cell line Jurkat Clone E6-1. In this sense, we assessed the function of caspase activation signaling in controlling the mitochondrial-mediated apoptosis. Thus, we found that hirsutine is capable of inducing mitochondrial apoptosis in Jurkat Clone E6-1 cells. Our research provides novel insights into hirsutine-mediated apoptosis and, moreover, indicates that hirsutine may function as a valuable chemotherapy compound for treating human T-cell leukemia.

# MATERIALS AND METHODS

## Cell lines, kits and reagents

Jurkat Clone E6 cells were from the American Type Culture Collection (ATCC, Manassas, VA, USA). Control THLE-2 hepatocytes and tubular epithelial HK2 cells from humans were collected from the Cell Resource Center of Shanghai Institutes for Biological Sciences, Chinese Academy of Sciences (Shanghai, China). RPMI-1640 medium and FBS were introduced from Hyclone Company (Cramlington, Northumberland, UK). Hirsutine (ST17300105, 5 mg/dose, purity ≥ 98%) was purchased from Shanghai Shidande Biotechnology Company (Shanghai, China). CCK8 kit, AnnexinV-FITC apoptosis detection kit, ECL chemiluminescence kit, RIPA Lysis Buffer and BCA protein assay kit were from Shanghai Beyotime Biotech (Shanghai, China). Cytoplasmic protein extraction kit was purchased from Invitrogen (Carlsbad, CA, USA). Antibodies against β-actin (BM0627) was purchased from BOSTER Biological Technology (Wuhan, China). Antibodies against Bcl-2 (12789-1-AP), Bax (50599-2-Ig) were from Proteintech Group (Wuhan, China), cleaved-caspase3 (Ab32042), Cyto C (Ab133504) were from abcam (Shanghai, China), cleaved-caspase9 (AF5240) was from Affinity (Shanghai, China). Other reagents were analytical reagent grade and from commercial sources.

## Cell culture

Human Jurkat Clone E6-1 cells were grown in RPMI-1640 containing 10% FBS and penicillin/streptomycin in a 5% $CO_2$ humidified incubator at 37 °C. Cells at 80% confluency were treated accordingly with hirsutine at different concentrations.

## Preparation of hirsutine solution

Hirsutine was dissolved in DMSO at the concentration of 100 mM and kept at −20 °C. The stock was diluted with RPMI-1640 medium to 10, 25 and 50 µM, respectively. Final DMSO concentration in working solution was kept at 0.1%. RPMI-1640 containing 0.1% DMSO was utilized for the untreated cell group.

## Cell viability analysis

Jurkat Clone E6-1 Cells were loaded into 96-well plate (10,000 cells/well) for 24 h. Thereafter, cells were treated or not with different hirsutine concentrations (10, 25 and 50 µM) for 48 h. Alternatively, plated cells were treated using increasing amounts of hirsutine (3.9, 4.7, 9.4, 18.8, 37.5, 75, 150 and 300 µM) for 24, 48 and 72 h. Normal human THLE-2 hepatocytes and normal human tubular epithelial cells HK2 were loaded into 96-well plate at 4,000 cells per well. Thereafter, normal human THLE-2 hepatocytes and normal human tubular epithelial cells HK2 were treated with different concentrations of hirsutine (15.625, 31.25, 62.5 and 125 µM) for 48 h. Afterwards, 10 µL of CCK8 reagent was dripped into per well, then preserving it in an incubator at 37 °C for extra 4 h. Spectrophometric measurement (absorbance at 450 nm) was then accessed per each well. Three replicates were analyzed for each cell treatment.

## Apoptosis detection by flow cytometry

Jurkat Clone E6-1 cells at logarithmic phase were plated into 6-well dishes at $1 \times 10^5$ cells/well for 24 h. Subsequently, cells were treated with increasing amounts of hirsutine (10, 25 and 50 µM) or 0.1% DMSO (negative control). Three biological replicates were assayed per condition. Cells were further collected and rinsed once with pre-cooled PBS. After cell resuspension using pre-cooled binding buffer, AnnexinV-FITC was added, gently mixed with cell suspension and incubated at room temperature for 15 min. Cells were then collected by centrifugation at 1,500 rpm for 5 min and, after discarding supernatant, cells were again resuspended in pre-cooled binding buffer. Thereafter, PI staining solution was added, mixed gently with cell suspension, and stored at 4 °C in the dark. Cells were immediately analyzed by flow cytometry (Becton, Dickinson and Company, Franklin Lakes, NJ, USA).

## Cell cycle distribution

Cells at the logarithmic proliferation period were plated into 6-well dish with $10^6$ cells/well for 48 h, and then distinct dosages of hirsutine were added in for another 48 h. Cells were then centrifuged at 1,000 rpm for 5 min. Precipitated cells were rinsed two times with cold PBS prior to fixation using 75% ethanol at 4 °C for at least 4 h. Thereafter, 400 µL
propidium iodide (PI, 50 μg/mL) and 100 μL RNase A (100 μg/mL) were added and cell suspension was incubated at 4 °C in the dark for 30 min. Cells were further evaluated by flow cytometry using standard procedures.

## Western blotting

Cells were grouped and treated as previously described. After 48 h of hirsutine treatment, the cells were lysed for 30 min in cold RIPA buffer (Beyotime, Haimen, China). The obtained lysates were spun for 15 min at 12,000 rpm at 4 °C, and the BCA method (Beyotime, Haimen, China) was then employed to quantify protein levels in supernatants. Next, 50 μg of protein in each sample was separated via SDS-PAGE and transferred to PVDF membranes. At room temperature, blots were blocked using 5% skim milk for 2 h prior to incubation along with different primary antibodies (1:1,000 dilution) at 4 °C overnight. Blots were further washed 3 times with 1xTBST, and probed for 1 h with secondary antibody (1:5,000) at room temperature. Blots were washed thrice using 1xTBST again, and the protein bands were visualized with an ECL system (Bio-Rad, Shanghai, China).

## QPCR analysis

Cells at the logarithmic phase were plated into 6-well dish at $10^6$ per well, cultured for 48 h, and then treated with hirsutine at different doses for 48 h. Total RNA was extracted with TRIpure Regent reagent (Ambion; Thermo-Fisher Scientific,Waltham, MA, USA) and digested by FastKing gDNA with RT SuperMix (Tiangen, Beijing, China) for reverse transcription. The expressions of mRNA of Bax, Bcl-2, cleavage-caspase 3, cleavage-caspase 9, Cyto c and GAPDH were detected by TaqMan probe. QPCR was carried out, in accordance with instruction manual, using ChamQ TM SYBR qPCR Master Mix and Applied Biosystems SDS 7500 instrument (Applied Biosystems Inc., Foster City, CA, USA). PCR reactions were submitted to the following cycling conditions: 50 °C for 2 min, 95 °C for 10 min and 40 cycles of 95 °C for 30 s, 60 °C for 30 s. PCR samples were loaded onto ethidium bromide-containing 2% agarose gels and analyzed by UV spectrophotometry. Primer sequences were as follows: bax (223 bp), F-5-GGCCCTTTT GCTTCAGGGTT-3, R-5-AGCTGCCACTCGGAAAAAGA-3, bcl-2 (383bp), F-5-GA CAACATCGCCCTGTGGAT-3, R-5-GACTTCACTTGTGGCCCAGAT-3, caspase-3 (182bp), F-5-TGGAACCAAAGATCATACATGGAA-3, R-5-TTCCCTGAGGTTTGC TGCAT-3, caspase-9 (193bp), F-5-AGGCCCCATATGATCGAGGA-3, R-5-TCGACAAC TTTGCTGCTTGC-3, gapdh (115bp), F-TCAAGAAGGTGGTGAAGC-AGG, R-TCAAA GGTGGAGGAGTGGGT.

## Statistical analysis

All the above data are means ± SD of three or more experiments. Differences between groups were compared through ANOVAs and $t$-tests. $P < 0.05$ was the significance threshold.

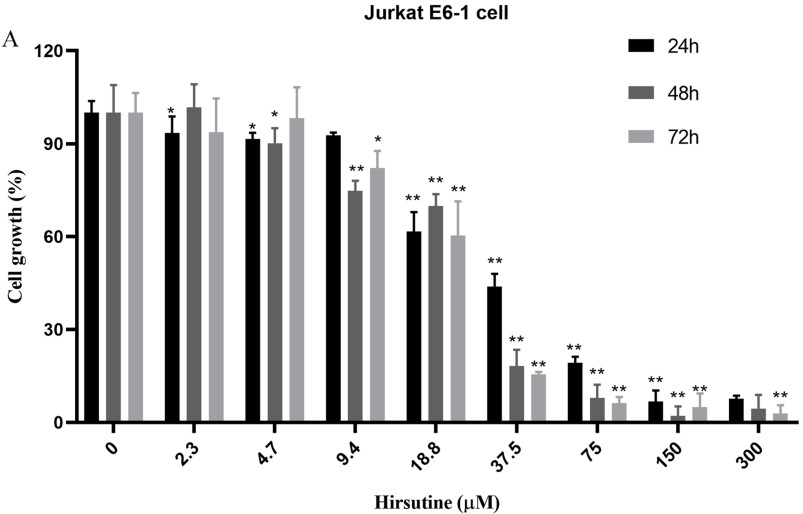

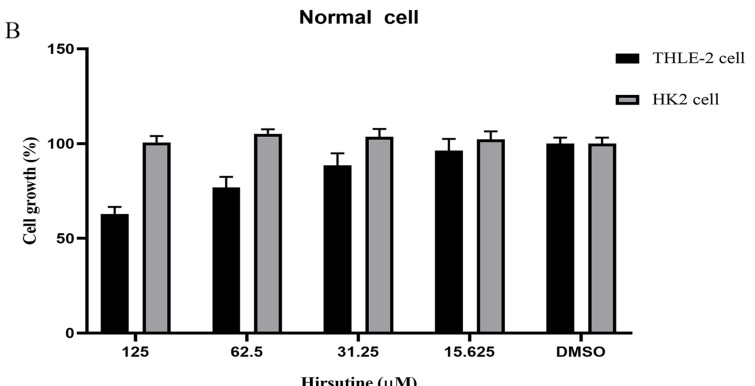

**Figure 2 Effect of hirsutine on Jurkat Clone E6-1 cell growth.** (A) Hirsutine inhibited human Jurkat Clone E6-1 cell line in vitro. Jurkat Clone E6-1 cell lines were treated with different doses of hirsutine for 24, 48, 72 h. Cell proliferation was determined using a CCK8 assay, $^*p < 0.05$, $^{**}p < 0.01$, (B) effect of hirsutine on normal cells survival. Cells were treated with different doses of hirsutine for 48 h. Cell survival was determined by CCK8 assay, $^*p < 0.05$, $^{**}p < 0.01$.

## RESULTS

### Impact of hirsutine on Jurkat clone E6-1 cell viability

Firstly, we investigated the impacts of hirsutine on cellular proliferation in human leukemia cells (Jurkat Clone E6-1), normal human THLE-2 hepatocytes and normal human tubular epithelial cells HK2. T-cell leukemia Jurkat Clone E6-1 cells were treated with increasing doses of hirsutine (3.9, 4.7, 9.4, 18.8, 37.5, 75, 150 and 300 µM) for 24, 48 and 72 h, respectively. Normal human THLE-2 hepatocytes and human normal tubular epithelial cells HK2 were treated with a range of hirsutine doses for 48 h. Exposure of Jurkat Clone E6-1 cells to hirsutine markedly impaired their survival in a dose and time-dependent fashion. Higher drug concentrations, used for a longer time of treatment, resulted in a more pronounced inhibitory effect on cells. In fact, more significant differences were observed at concentrations higher than 37.5 µM (Fig. 2A). However, in

case of normal human THLE-2 hepatocytes and normal human tubular epithelial cells HK2, after 48 h of drug exposure, hirsutine had nearly no influence (Fig. 2B). These data suggest that hirsutine effectively and selectively inhibit Jurkat Clone E6-1 cell proliferation (Fig. 2).

## Effect of hirsutine on the apoptosis cell death of Jurkat E6-1 cells

To examine whether the cytotoxic activity of hirsutine is linked to apoptotic cell death, Jurkat E6-1 cells, were treated with hirsutine and Annexin V-FITC staining assay was performed. As shown in Fig. 3, after 48 h of treatment with 10, 25 and 50 µM hirsutine, the percentage of Annexin V-FITC positive cells increased up to 4.99 ± 0.51%, 13.69 ± 2.00% and 40.21 ± 15.19% in Jurkat cells, respectively. These results suggest that the cell growth inhibitory effect of hirsutine is linked to apoptotic death in human Jurkat Clone E6-1 cells.

## Hirsutine induces G0/G1 phase arrest

Cell cycle arrest is an additional mechanism that can disrupt the growth of tumor cells (Qiu et al., 2011). In order to explore how hirsutine impacts cellular proliferation, we examined the inhibition of such proliferation was a consequence of cell cycle arrest. Treatment with hirsutine was related to a considerable increase in G0/G1 phase Jurkat cells (Fig. 4). In the control group, normal cells distributed in the G0/G1 period accounted for 20.54 ± 4.23% of the total cell population and those of stage S and G2/M each accounted for 78.23 ± 3.13% and 1.26 ± 1.20%. However, after treatment with 50 µM hirsutine for 48 h, the percentage of cells in G0/G1 phase rose to 49.12 ± 4.07% significantly, and decreased in phase S and G2/M, which shows each doses have statistically significant, compared with the control ($P < 0.05$ or $P < 0.01$) (Fig. 4). Overall, our results demonstrate that hirsutine might inhibit cell viability of Jurkat Clone E6-1 cells via inducing G0/G1 phase cell arrest, suggesting that the cell growth inhibition effect of hirsutine was mediated by cell cycle control (Fig. 4).

## Hirsutine induces the mitochondrial dysfunction in human Jurkat clone E6-1 cells

To elucidate whether hirsutine drives apoptotic death via the induction of the mitochondrial pathway of apoptotic cell death, we quantified changes in Bcl-2 expression and caspase activation. Mitochondrial membrane potential changes can be used to monitor abnormal mitochondrial functionality early during apoptosis. Immunoblotting revealed that Jurkat Clone E6-1 cell treatment with hirsutine increased Bax, cleaved-caspase 3/9 and Cyto c protein levels (Fig. 5A). However, hirsutine was associated with reductions in Bcl-2 expression, resulting in a rise in the Bcl-2 pro-/anti-apoptotic ratio (Figs. 5B–5D). Apoptotic progression is characterized by the activation of caspases that control the cell death cascade. Hirsutine enhanced upstream caspase-9 activity in Jurkat Clone E6-1 cells (Fig. 5E). In addition, hirsutine also increased the activation of effector caspase-3 in Jurkat cells (Fig. 5F). Moreover, hirsutine drove mitochondrial

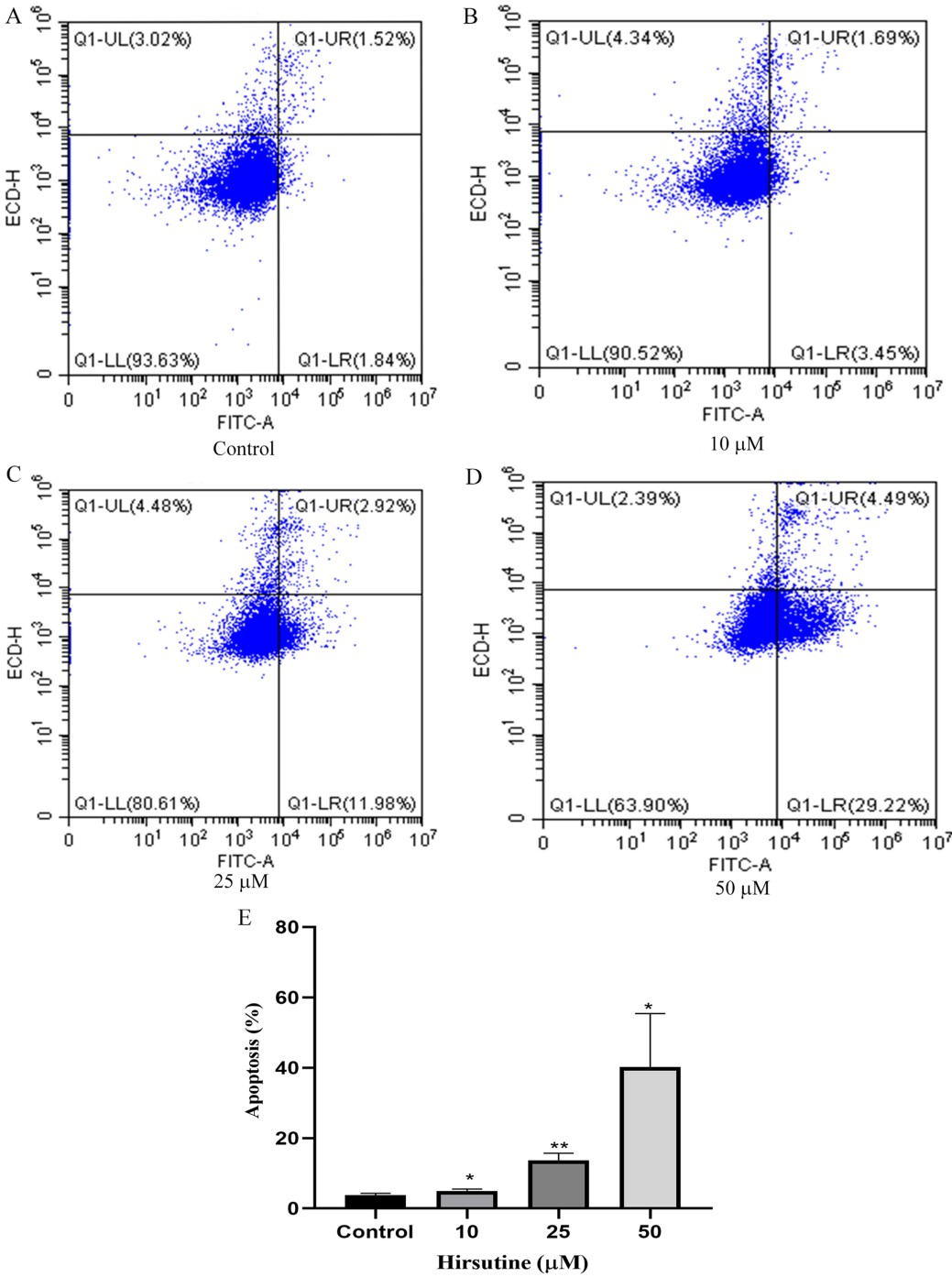

**Figure 3 Apoptosis of Jurkat Clone E6-1 cells after treatment with different doses of hirsutine.** Jurkat Clone E6-1 cells were treated with different doses of hirsutine for 48 h and measured by flow cytometry. (A) Control, (B) 10 µM, (C) 25 µM, (D) 50 µM, (E) histogram of apoptosis of Jurkat Clone E6-1 cells, $*p < 0.05$, $**p < 0.01$ ($n = 3$).

Cytochrome c release into the cytosol in a dose-dependent fashion (Fig. 5G). It is further confirmed that hirsutine induces apoptosis of human Jurkat clone E6-1 cells by promoting mitochondrial dysfunction.

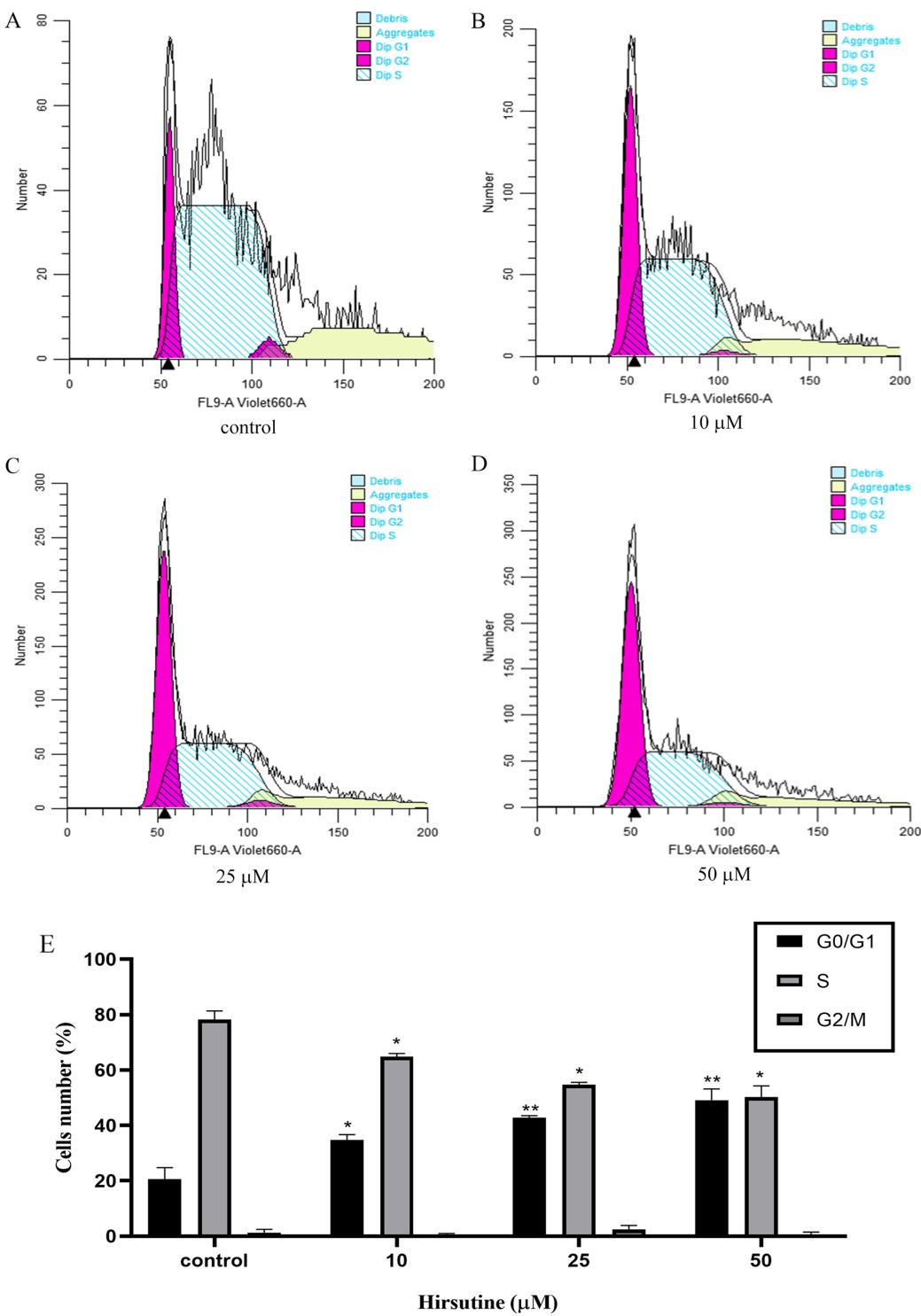

**Figure 4 Cell cycle of Jurkat Clone E6-1 cells after treatment with different doses of hirsutine. Jurkat Clone E6-1 cells were treated with different doses of hirsutine for 48 h and the cell cycle were measured by flow cytometry.** (A) Control, (B) 10 µM, (C) 25 µM, (D) 50 µM, (E) histogram of cell cycle of Jurkat Clone E6-1 cells, $^*p < 0.05$, $^{**}p < 0.01$ ($n = 3$).

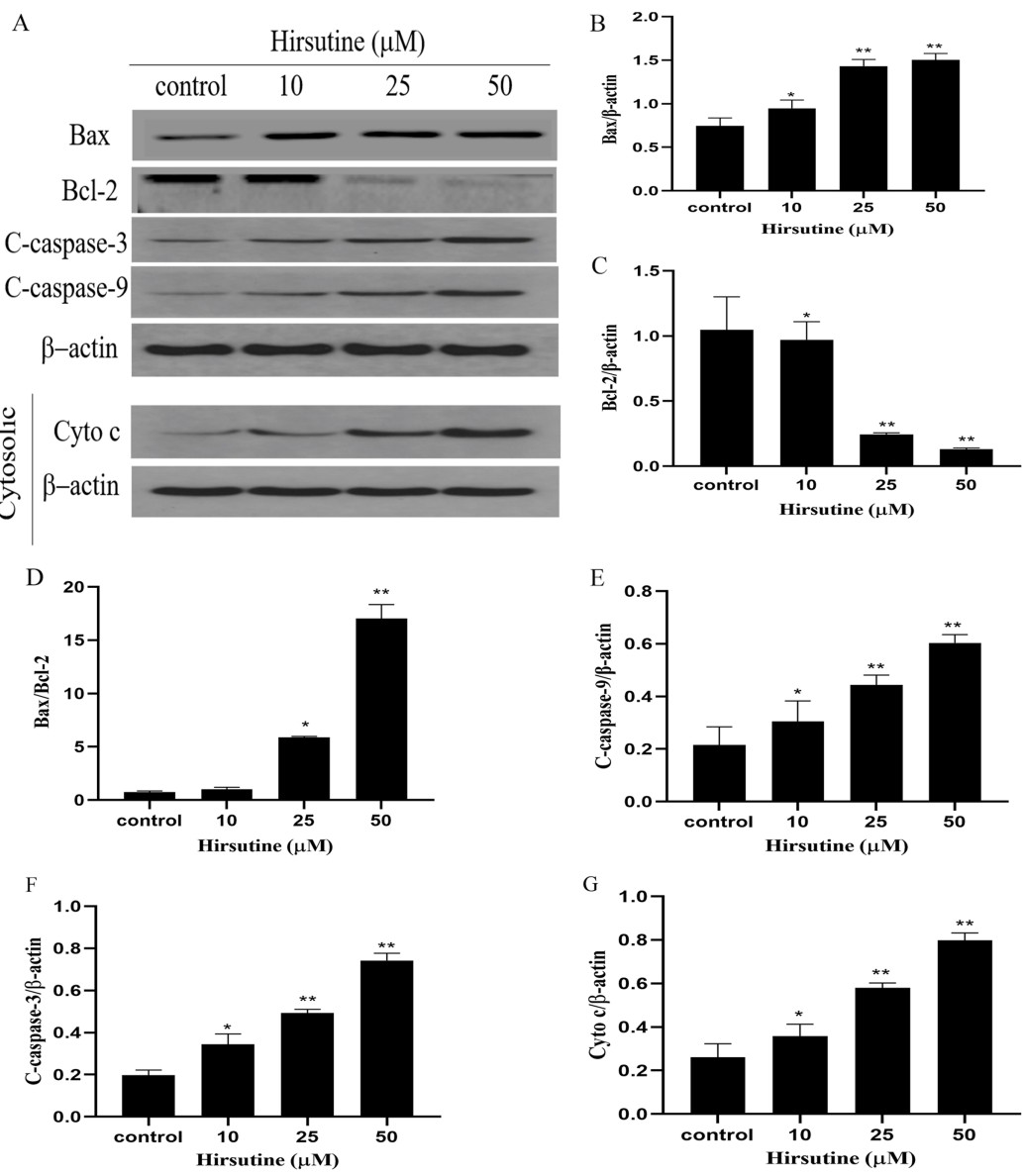

**Figure 5 Effects of hirsutine on Bcl-2 family proteins, and caspase activation proteins in Jurkat Clone E6-1 cells.** After 48 h treatment by different doses of hirsutine, the bax, bcl-2, cleaved-caspase-3 (C-Caspase 3), cleaved-caspase-9 (C-Caspas 9) and cytochrome c (Cyto c) in the Jurkat Clone E6-1 cells were measured by Western blotting. (A) Total cellular extracts and cytosolic fractions were analyzed by Western blot analysis using antibodies against Bax, Bcl-2, cleaved-caspase-3 (C-Caspase 3), cleaved-caspase-9 (C-Caspase 9), and cytochrome c (Cyto c), (B) bax protein content of Jurkat Clone E6-1 cells in different doses of hirsutine, (C) bcl-2 protein content of Jurkat Clone E6-1 cells in different doses of hirsutine, (D) histogram of bax/bcl-2 ratio, (E) cleaved-caspase-9 (C-Caspase 9) protein content of Jurkat Clone E6-1 cells in different doses of hirsutine, (F) cleaved-caspase-3 (C-Caspase 3) protein content of Jurkat Clone E6-1 cells in different doses of hirsutine, (G) cytochrome c (Cyto c) protein content of Jurkat Clone E6-1 cells in different doses of hirsutine, $^*p < 0.05$, $^{**}p < 0.01$ ($n = 3$).

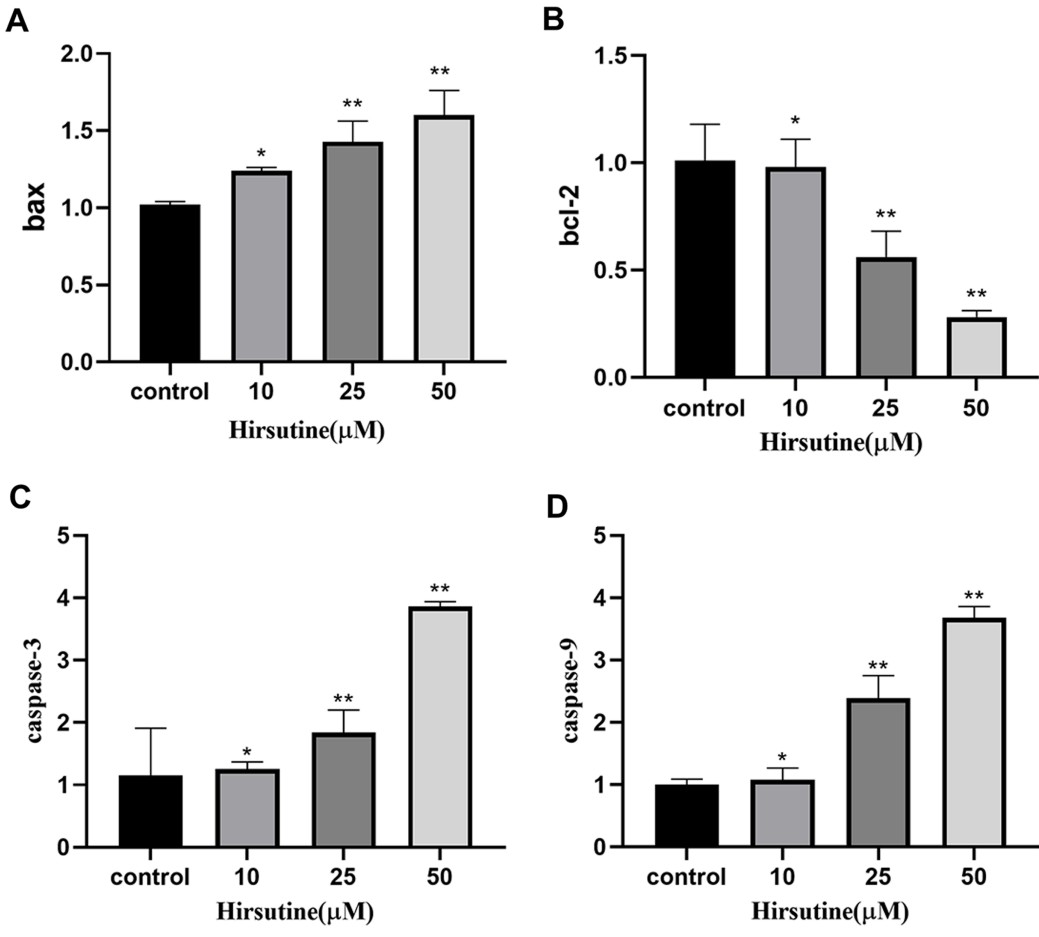

**Figure 6 The bax/bcl2, caspase-3/9 mRNA in Jurkat Clone E6-1 cells after treatment with different doses of hirsutine.** After 48 h treatment by different doses of hirsutine, bax/bcl2, caspase-3/9 mRNA in the Jurkat Clone E6-1 cells were measured by qPCR. (A) Histogram of bax, (B) histogram of bcl-2, (C) histogram of caspase-3, (D) histogram of caspase-9, $*p < 0.05$, $**p < 0.01$ ($n = 3$).

## Changes on the levels of Bcl-2, Bax, caspase-3 and caspase-9 mRNA

Quantitative polymerase chain reaction analysis showed that, after 48 h of treatment with different doses of hirsutine, Bax mRNA levels increased, while *BCL2* expression decreased. The mRNA levels of caspase-3 and caspase-9 were also increased in hirsutine-treated Jurkat cells. The concentrations of 10, 25, and 50 μM hirsutine were more effective than the untreated group ($p < 0.05$) (Fig. 6).

## DISCUSSION

Research on new chemotherapeutics is inseparable from TCM, because it is still suitable for storage of new molecules (*Yun et al., 2012*; *Zheng et al., 2017*). Hirsutine, an indole alkaloid, originally isolated from *U. rhynchophylla*, has an anti-cancer property, whose efficacy rises relying on the dosage and hours in vivo (*Huang et al., 2018*; *Zhang et al., 2018*). Through this experiment, we have confirmed that hirsutine can inhibit the growth of Jurkat Clone E6-1 cells and promote cell death to a certain extent. Specifically, it can

stop cell growth in G0/G1 phase, and cause cell death through ajusting caspase activation signaling pathway and cell-cycle regulatory proteins.

The antitumor activity of hirsutine has been well reported (*Huang et al., 2018*; *Zhang et al., 2018*). First, we evaluated the effects of hirsutine on the proliferation of human leukemia cells (Jurkat Clone E6-1), normal human THLE-2 hepatocytes and normal human tubular epithelial cells HK2 cells. After administration, different incubation time corresponds to different cell proliferation cycle numbers, and the determination of action time is related to the growth cycle of tumor cells. The reason why the drug acts on cells for 48 h also depends on the number of cells at that time. Cells enter the exponential growth stage after 12–24 h of passage, and enter the plateau stage 48–72 h after passage generally. At this time, the administration effect is the most obvious and typical. T-cell leukemia Jurkat Clone E6-1 cells were dealed with increasing dosages of hirsutine for 24, 48 and 72 h, respectively. Normal human THLE-2 hepatocytes and normal human tubular epithelial cells HK2 were dealed with various dosages of hirsutine for 48 h. After treated with hirsutine, the cell viability of Jurkat clone E6-1 cells decreased significantly under conditions of different concentrations and hours (Fig. 2). However, despite being treated with hirsutine for 48 h, THLE-2 hepatocytes and normal human renal tubular epithelial cells HK2 (Figs. 2A and 2B) were almost not been inhibited. These data suggest that hirsutine has a selectively inhibitory effect on the growth of Jurkat cells.

Mitochondria are closely linked to apoptotic induction, and proteins in the Bcl-2 family regulate the function of these organelles, including both pro-apoptotic Bax and anti-apoptotic Bcl-2 (*Adams & Cory, 2007*; *Hengartner, 2000*). In this study, the increased expression of Bax and Cyt c indicated that hirsutine (10, 25, and 50 µmol/L) can promote the release of Cyt c from mitochondria, thereby inducing the initiation of mitochondrial-mediated pathways. As an anti-apoptotic protein, bcl-2 can prevent the release of Cyt c from mitochondria (*Liu et al., 2013*). Some studies have shown that hirsutine induces apoptosis by down-regulating the expression of Bcl-2 (*Huang et al., 2018*; *Zhang et al., 2018*). In this study, we focused on the mitochondrial-dependent pathway to cell death. The up-regulation of BCL2 levels may be the compensatory protective effect of hirsutine on Jurkat Clone E6-1 cell apoptosis at concentrations of 10, 25 and 50 µmol/L. Hirsutine treatment led to marked reductions in Bcl-2 expression and enhanced Bax expression, suggesting that this shift in the Bax/Bcl-2 ratio may govern the apoptotic function of hirsutine. Moreover, pro-apoptotic Bax can form pores in the mitochondrial membrane and facilitate cytochrome c release (*Antonsson et al., 1997*). The cells released into the cytoplasm can promote the binding of Apaf-1 and caspase-9 and promote the activation of caspase-9 (*Brentnall et al., 2013*; *Kole, Knight & Deshmukh, 2011*). Caspase family is an important class of proteases typically involved in the apoptosis pathway. Caspase-3, as a key factor in the execution of apoptosis, can be activated by activated capsase-9 (*Thomas et al., 2017*). Previous studies reported that hirsutine induces apoptosis by activating caspase-9 and caspase-3 (*Huang et al., 2018*; *Zhang et al., 2018*). In this study, the increased expression of Caspase-3 and Caspase-9 indicated that Caspase-3 and Caspase-9 caused the caspase cascade, which led to cell death. In short,

excessive hirsutine can destroy the ultrastructure of mitochondrial cells, leading to the release of mitochondrial Cytc and the initiation of mitochondrial pathways. In addition, Cyt c may activate the downstream cascade of caspases. Generally, excessive hirsutine can induce apoptosis of Jurkat Clone E6-1 cells through a mitochondrial-mediated pathway.

Inducing cell cycle arrest is a primary antitumor treatment strategy (Asci Celik et al., 2020). Therefore, elucidating the way in which hirsutine inhibits cell cycle progression may provide a mechanism basis for the anticancer effects of these herbs. Although previous studies have shown that it is difficult to determine the target of hirsutine, this study shows that hirsutine blocks the cell cycle in G0/G1 phase, that is to say, it blocks the cell cycle at the early stage of DNA synthesis, prevents the synthesis of RNA and protein at this stage, and prevents the cell cycle from entering the S phase. The results showed that hirsutine blocked tumor cell proliferation via inducing G0/G1 phase arrest.

Bcl-2 family proteins are regulatory factors of apoptosis. Many pro-apoptotic members of this family, such as Bax and Bak, govern caspase-mediated cell death pathway. The ratio of Bcl-2/Bax protein may explain the protective mechanism of Bcl-2 protein in cells. Our results indicate that hirsutine disrupts the balance of pro- and anti-apoptotic proteins in the Bcl-2 family, leading to the destruction of mitochondrial membrane and intrinsic pathway mediated apoptosis. QPCR analyses were further performed to evaluate mRNA levels of Bax, Bcl-2, caspase-3 and caspase-9. As previously indicated, hirsutine could diminish Bcl-2 mRNA levels and, at the same time, improve Bax, caspase-3 and caspase-9 mRNA content, thus increasing the ratio of pro-versus anti-apoptotic proteins.

## CONCLUSIONS

In the present work, we show that an important bioactive component isolated from *U. rhynchophylla*—hirsutine—had inhibitory effect on Jurkat Clone E6-1 cells, because it can promote cell proliferation in the G0/G1 phase, and inhibit cell growth in the S and G2/M phase, promoting cell death upon elevating Bax, cleaved-caspase 3/9 and Cyto c proteins but decreasing the yields of Bcl-2 protein. At the same time, hirsutine treatment also elevated caspase-3 and caspase-9 mRNA levels, suggesting that hirsutine has a potential antitumor activity, thus enlightening the use of phytomedicines in tumor therapy.

In summary, the research has profoundly investigated the antitumor effects of hirsutine, showing its possible effects on treating T-cell leukemia. More in-depth studies will be badly needed to explore more precise mechanisms related to the hirsutine-mediated antitumor activity.

## ACKNOWLEDGEMENTS

The authors thank the reviewers for their helpful comments on this report.

### Funding

The authors received no funding for this work.

## Competing Interests

The authors declare that they have no competing interests.

## Author Contributions

- Jie Meng conceived and designed the experiments, performed the experiments, analyzed the data, prepared figures and/or tables, and approved the final draft.
- Rui Su performed the experiments, analyzed the data, authored or reviewed drafts of the paper, and approved the final draft.
- Luping Wang analyzed the data, prepared figures and/or tables, authored or reviewed drafts of the paper, and approved the final draft.
- Bo Yuan analyzed the data, authored or reviewed drafts of the paper, and approved the final draft.
- Ling Li conceived and designed the experiments, prepared figures and/or tables, and approved the final draft.

## Data Availability

The raw measurements are available in the Supplemental Files.

## Supplemental Information

Supplemental information for this article can be found online at http://dx.doi.org/10.7717/peerj.10692#supplemental-information.

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
