# Peer review of "Inhibitory effect and mechanism of action (MOA) of hirsutine on the proliferation of T-cell leukemia Jurkat clone E6-1 cells"

_PeerJ, doi:10.7717/peerj.10692_

## Round 0.1 · original submission · Major Revisions

Please take special care in responding to reviewer #1, whose criticism seems to me to be the most important.

Reviewer 1 ·

Basic reporting

The manuscript requires English language editing. The introduction is ambiguous not providing a direct focus of the aim of the study. The results and raw results are presented. However, there are issues with the figures, figure 5 and 6 do not have units. Usually, in RT PCR there is a ratio with GADPH or actin. The discussion is poor and requires rewriting

Experimental design

The experimental design lacks important details. The Jurkat cells were obtained from? What were the conditions of cell growth in culture?, ie, how do we know that the cells were not overgrown? There is 5 % necrosis along with 3.6 % late apoptosis, in total around 9 % of the cells were IP positive in the control, it is too high. Why not perform the assays with a range of concentrations from nanoM to microM? The assays with bcl are not performed correctly, probably cytochrome c release would have been a better option. The cell cycle experiments look ok in principle, but the results suggest that the cells were overgrown. The RT PCR data lacks details of the number of cells used.

Validity of the findings

There seems to be an effect of hirsutine, but the results are not being presented in the adequate form. The cell culture should be synchronized in order to ascertain the effect on apoptosis which is moderate, and some assays like RT PCR or bcl are not very convincing.

Additional comments

Please restructure the paper. Perform experiments with cells with less than 2 % IP in the controls, perform curves with the compound from nanoM to microM, either assess caspase activity or cytochrome c release from the mitochondria. Finally, the introduction and the discussion should be rewritten with specific aims and direct conclusions. English language editing is required.

·

Basic reporting

The English is clear.

The references are sufficient.

The Figure 2 is necessary changed because the asterisks are not in the correct position

It is important to precise the innocuity because in this manuscript there are not experiments to confirm the not toxicity.

Experimental design

The aims are clear
The research question is clear and the methods to answer correct.
Methods was described with sufficient detail to be replicate.

Validity of the findings

The results obtained are important to considerer the hirsutine as probably leukemia treatment.
I suggest check the statistical analysis on the first to bars of the figure 5 bax/bcl-2.
To improve the abstract, I suggest including the concentrations studied.
Conclusions are well stated supported by the results.

Additional comments

The manuscript need minor reviisions in order to improve the article.
I think it is important to pay attention to figure 2.

Reviewer 3 ·

Basic reporting

The hypothesis modeled by the authors is well clear; the methods are proper and well defined; the data are complete; the manuscript adheres to the relevant standards for reporting and data deposition

Experimental design

Appropriate research techniques and data analysis were employed

Validity of the findings

All underlying data have been provided; they are robust, statistically sound, & controlled.
Conclusions are well stated, linked to original research question & limited to supporting results.

Additional comments

Jie Meng et al investigated the antitumour activity of hirsutine isolated which is isolated from Uncaria rhynchophylla and its mechanism of action. The scope of the manuscript is clearly defined, and the title is adequately interpreted. The abstract presented the aims, objectives, methods, results as well as conclusion adequately. Introduction, there is strong evidence of engagement with relevant and current literature, and all the references listed were cited in the text/body of the thesis. Appropriate research techniques and data analysis were employed, and the manuscript is highly structured and argued. However few questions need to be addressed.

What is the source of hirsutine? What is its purity?
What is the role of free radicals in hirsutine-induced apoptosis?
What would be the effect in normal cells at the tested concentration of cancer cells?
The apoptosis induction is mitochondrial dependent or independent?
In general, activation..i.p..cleavage of caspases are more important than the gene expression. Authors may repeat the experiment with western blotting using cleaved caspases.
It would be better to include antibody catalog number in the methodology
Gene accession number may be included in the methodology.
Discussion part needs to be strengthened

---

## Round 0.2 · Major Revisions

There are still many issues remaining. Please address them thoroughly.

Reviewer 1 ·

Basic reporting

The manuscript has improved partially. There are some changes which were suggested and not performed. The English language revision is still required. As an example in the new sentence in the abstract of the manuscript states, Western blotting were utilized to discover the expression, which is not scientifically correct. There are may other grammatical mistakes in the corrected manuscript.
The experimental design was modified, but still, there are gaps in how part of the cell cycle experiments were performed.
Finally, the discussion was modified but it is still lacking a proper structure. The effects of hirsutin are after 48 hrs of culture at micro Molar concentrations suggesting that the compound is unable to directly affect cell viability. Thus, cell control mechanisms should be discussed.

Experimental design

The authors made most of the required changes, but still, a description of cell cycle analysis is not detail as requested

Validity of the findings

Most of the crucial issues remain unsolved. Certainly, the authors have data pointing out the effect of the compound, but the data still not well presented. First, the most important element is to ascertain the effect of the compound intrinsic of the extrinsic apoptotic pathways which were done poorly. The effect is observed at 48 hr at mico M concentrations. Figure 3 there is a conflict between the flow cytometry assessment, performed poorly, and the figure which the percentages do not match. Figure 4, there is a clear decrease in the DNA density; however, the interpretation in the figure below is not correct. Figure 5 was improved showing the importance of cytochrome c, but the results on caspase 3 and 9 are very similar. In figure 6, based on the previous data, it is impossible that the values of caspase 3 and 6 are the same upon treatment. Thus, the authors claim a decrease in chromatin structure based on mitochondrial induced apoptosis but show that the extrinsic apoptotic induction is also present without even analyzing CD95. The results do not support this conclusion.

Additional comments

The paper is still lacking the conditions to be published. It still requires mejor revision.

---

## Round 0.3 · Minor Revisions

Please attend to the few remaining minor issues.

Reviewer 1 ·

Basic reporting

The article was improved significantly. The use of the English language has been improved as well as other important points

Experimental design

Experimental design now is more understandable and is according to the journal guidelines.

Validity of the findings

The findings are better presented and sicussed.

Additional comments

The manuscript is highly improved with all the detailed responses.

·

Basic reporting

The English is clear and the references enough, The figures is necessary to improve the figure 6 in order to precise if it is RT_PCR or qPCR.
The results are important and could be useful in the leukemia treatment.

Experimental design

The aims of the manuscript is clear and the experimental design is ok, the question is clear in order to know if hisutine has effect on the leukemia cells studied and the results important to think in the use of this molecule in the treatment.

Validity of the findings

The results are important and novelty for leukemia, the data are robust but it is important to precise the type of PCR used, the conclusions are well stated because they used as control a normal cells.

Additional comments

I will like to comment about the basic report: the manuscript has been write in clear English and have sufficient references. The experimental design is good and the results importan to think in the possibility to add hirsutine on the leukemia treatment.
In order to improve the manuscript I suggest introduce a reference about the hirsutine safety because it has been used in others sickens.
A main point to attend is to precede in abstract, methods, results , discussion and figure if the PCR used is qPCR, because now is confuse.
Warm regards
Guadalupe

---

## Round 0.4 · accepted · Accept

Thank you for the latest changes.